# The inhibitory effects of toothpaste and mouthwash ingredients on the interaction between the SARS-CoV-2 spike protein and ACE2, and the protease activity of TMPRSS2 *in vitro*

**Riho Tateyama-Makino**[1‡]*, **Mari Abe-Yutori**[1‡], **Taku Iwamoto**[1], **Kota Tsutsumi**[1], **Motonori Tsuji**[2], **Satoru Morishita**[1], **Kei Kurita**[1], **Yukio Yamamoto**[1], **Eiji Nishinaga**[1], **Keiichi Tsukinoki**[3]

1 Research & Development Headquarters, Lion Corporation, Edogawa-ku, Tokyo, Japan, 2 Institute of Molecular Function, Misato-shi, Saitama, Japan, 3 Division of Environmental Pathology, Department of Oral Science, Kanagawa Dental University, Yokosuka, Kanagawa, Japan

‡ These authors have contributed equally to this work and share first authorship
* riho-t@lion.co.jp

## Abstract

SARS-CoV-2 enters host cells when the viral spike protein is cleaved by transmembrane protease serine 2 (TMPRSS2) after binding to the host angiotensin-converting enzyme 2 (ACE2). Since ACE2 and TMPRSS2 are expressed in the tongue and gingival mucosa, the oral cavity is a potential entry point for SARS-CoV-2. This study evaluated the inhibitory effects of general ingredients of toothpastes and mouthwashes on the spike protein-ACE2 interaction and the TMPRSS2 protease activity using an *in vitro* assay. Both assays detected inhibitory effects of sodium tetradecene sulfonate, sodium N-lauroyl-N-methyltaurate, sodium N-lauroylsarcosinate, sodium dodecyl sulfate, and copper gluconate. Molecular docking simulations suggested that these ingredients could bind to inhibitor-binding site of ACE2. Furthermore, tranexamic acid exerted inhibitory effects on TMPRSS2 protease activity. Our findings suggest that these toothpaste and mouthwash ingredients could help prevent SARS-CoV-2 infection.

## Introduction

Coronavirus disease 2019 (COVID-19), which is caused by severe acute respiratory syndrome coronavirus 2 (SARS-CoV-2), is a serious public health problem worldwide. SARS-CoV-2 mainly infects the upper respiratory tract, which leads to respiratory disease. The viral spike protein mediates SARS-CoV-2 entry into host cells [1]. To fulfill this function, it is essential that the spike protein bind to angiotensin-converting enzyme 2 (ACE2) on the host cells [1]. The transmembrane protease serine 2 (TMPRSS2) of the host is an essential factor of SARS--CoV-2 entry into host cells [2]. TMPRSS2 cleavages the spike protein bound to ACE2, and the cleaved spike protein leads to the fusion of viral and host cell membranes [2]. Therefore, the

**Data Availability Statement:** All relevant data are within the paper and its Supporting Information files.

**Funding:** This work was fully funded by Lion Corporation (https://www.lion.co.jp/en/). R Tateyama-Makino, M Abe-Yutori, T Iwamoto, K Tsutsumi, S Morishita, K Kurita, Y Yamamoto, and E Nishinaga are employees of Lion Corporation. The Lion Corporation provided support in the form of salaries for authors R Tateyama-Makino, M Abe-Yutori, T Iwamoto, K Tsutsumi, S Morishita, K Kurita, Y Yamamoto, and E Nishinaga. M Tsuji is a President of the Institute of Molecular Function (Saitama, Japan). Molecular docking simulation was performed by the Institute of Molecular Function under a consignment from the Lion Corporation. K Tsukinoki has received fees for technical guidance from the Lion Corporation. The funders had no role in study design, data collection and analysis, decision to publish, or preparation of the manuscript. The specific roles of these authors are articulated in the "author contributions" section.

**Competing interests:** We have read the journal's policy and the authors of this manuscript have the following competing interests: R Tateyama-Makino, M Abe-Yutori, T Iwamoto, K Tsutsumi, S Morishita, K Kurita, Y Yamamoto, and E Nishinaga are employees of the Lion Corporation (Tokyo, Japan). M Tsuji is a President of the Institute of Molecular Function (Saitama, Japan). Molecular docking simulation was performed by the Institute of Molecular Function under a consignment from the Lion Corporation. K Tsukinoki has received fees for technical guidance from the Lion Corporation. This does not alter our adherence to PLOS ONE policies on sharing data and materials.

focus has been on finding agents that inhibit the viral spike protein-ACE2 interaction and/or TMPRSS2 serine protease activity to prevent SARS-CoV-2 infection [3, 4].

The oral cavity is an important entry point for pathogens. Xu et al. demonstrated that ACE2 was expressed in oral mucosa, especially the dorsal tongue [5]. Sakaguchi et al. and Huang et al. showed that ACE2 and TMPRSS2 are expressed in the mucosa of the tongue, gingiva, and salivary gland epithelial cells [6, 7]. These findings suggest that multiple oral epithelial cells serve as an entry point for SARS-CoV-2. Although whether the activities of ACE2 and TMPRSS2 on the tongue are associated with virus infection is unclear, taste impairment has been recognized as a symptom of COVID-19 [8]. Huang et al. suggested that SARS-CoV-2 infects the salivary glands and oral mucosa and implicates saliva in viral transmission [6]. These findings support that the oral cavity is an entry point for SARS-CoV-2. Hence, it is important to prevent SARS-CoV-2 infection by reducing the activity of the virus itself and inhibiting the entry point of the virus in the oral cavity. Carrouel et al. (2021) suggested that some ingredients of mouthwashes reduce the activity of the virus [9]. Povidone-iodine and surfactants, including those found in toothpastes and mouthwashes, are known to inactivate enveloped viruses, including SARS-CoV-2 [9, 10]. To further enhance the inhibitory effect of these ingredients on the entry of SARS-CoV-2 into host cells, it is also important to consider its effect on the virus' infection mechanism on the host cell side. However, the effects of ingredients of toothpastes and mouthwashes on the entry point of SARS-CoV-2 into host cells are unclear. Therefore, our study focused on the effects of general ingredients contained in commercially available toothpastes and mouthwashes on the entry of SARS-CoV-2 into host cells.

In this study, we evaluated the inhibitory effects of these ingredients on the interaction between the spike protein of SARS-CoV-2 and human ACE2 as well as the protease activity of TMPRSS2, *in vitro*. We also performed docking simulations for these target proteins to support the experimental results. As a result, we found that some general ingredients contained in commercially available toothpastes and mouthwashes exhibit inhibitory effects on the SARS-CoV-2 spike protein-ACE2 interaction and on the TMPRSS2 serine protease activity. These findings suggest that the ingredients in toothpastes and mouthwashes can possibly help prevent SARS-CoV-2 infection.

## Materials and methods

### Materials

The test ingredients (Table 1) were purchased from FUJIFILM Wako Pure Chemical Corporation, Nikko Chemicals Co., Ltd., and Lion Specialty Chemicals Co., Ltd. Each stock solution of test ingredients was diluted with phosphate-buffered saline (PBS) at a concentration of 5% (w/w). Recombinant human TMPRSS2 (N-terminus 6xHis, aa106-492) was purchased from LifeSpan Biosciences (Seattle, WA, USA). Boc-Gln-Ala-Arg-MCA was purchased from Peptide Institute Inc. (Osaka, Japan).

### *In vitro* assay of the interaction between receptor-binding domain of spike protein and ACE2

The interaction between the receptor-binding domain (RBD) of the SARS-CoV-2 spike protein and ACE2 was estimated using Spike S1 (SARS-CoV-2): ACE2 Inhibitor Screening Colorimetric Assay Kit (BPS Bioscience, San Diego, CA, USA). Test ingredients were added to a 96-well plate coated with Spike S1 and incubated for 1 hour. Subsequently, ACE2-biotin solution was added to the wells, followed by incubation for 1 hour. After washing the plate, streptavidin-HRP was added and incubated for 1 hour. After washing the plate again, the HRP

**Table 1. List of test ingredients used for the *in vitro* screening.**

| Purpose | Group | Abbreviation | Ingredient |
|---|---|---|---|
| Active Ingredient | Fluoride | MPS | Sodium monofluorophosphate |
| | | FLS | Sodium fluoride |
| | Amino acid | TXA | Tranexamic acid |
| | | AHA | 6-aminohexanoic acid |
| | Nitrate | PNI | Potassium nitrate |
| | Phosphate | TPS | Sodium tripolyphosphate |
| | | PPS | Sodium pyrophosphate |
| | Surfactant (Cationic) | CPC | Cetylpyridinium chloride |
| | | BZC | Benzyldimethyltetradecylammonium chloride |
| | Surfactant (Anionic) | LSS | Sodium N-lauroylsarcosinate |
| Foaming Agent | Surfactant (Anionic) | SDS | Sodium dodecyl sulfate |
| | | LMT | Sodium N-lauroyl-N-methyltaurate |
| | | TDS | Sodium tetradecene sulfonate |
| | Surfactant (Nonionic) | PEC | Polyoxyethylene (15) cetyl ether |
| | | PEG | Polyethylene glycol 4000 |
| | | GFE | Glycerin fatty acid ester |
| | | POE(20) | PEG-20 hydrogenated castor oil |
| | | POE(100) | PEG-100 hydrogenated castor oil |
| | Surfactant (Amphoteric) | SCP | Sodium cocoamphoacetate (40% solution) |
| | | COB | Cocamidopropyl betaine (30% solution) |
| Humectant | Sugar alcohol | PGL | Propylene glycol |
| | | GLY | Glycerol (85% solution) |
| | | XYL | Xylitol |
| | | SOR | Sorbitol (70% solution) |
| | Artificial sweetener | SAC | Sodium saccharin |
| Other | Amino acid | PCA | Sodium DL-pyrrolidonecarboxylate (50% solution) |
| | Phospholipid | GPC | Calcium 2-glycerophosphate |
| | Gluconate | GCU | Copper gluconate |
| | Amino acid | ALA | DL-alanine |
| | Citrate | CIS | Sodium citrate |

substrate was added, and the absorbance of the solution was measured. The absorbance value (AV) at 450 nm was read using Infinite 200 Pro (Tecan Japan, Kanagawa, Japan). All assays were performed, following the manufacturer's instructions. The inhibitory rate was calculated as follows: Inhibition (%) = (AV of control–AV of ingredients treatment) / AV of control × 100. The dose–response relationship between inhibition (%) and test ingredient concentration was plotted and used to determine the half maximal inhibitory concentration (IC50) using the DRC package in R software program (v3.6.1), as described in [11]. In brief, the dose–response data were fitted using a four-parametric log-logistic model or a four-parameter Brain-Cousens model when the dose–response data represented a sigmoid curve or hormesis, respectively. In addition, the obtained data was used for the estimation of the IC50.

### *In vitro* assay of TMPRSS2 serine protease activity

The serine protease activity of TMPRSS2 was estimated as previously described [12, 13]. Recombinant human TMPRSS2 (4 μg/mL final concentration) diluted with an assay buffer (50 mM Tris-HCl pH 8.0, 154 mM NaCl) and test ingredients were added to the 384 well black

plate (Greiner Bio-One Japan, Tokyo, Japan). Then, Boc-Gln-Ala-Arg-MCA (10 μM final concentration) diluted with an assay buffer containing dimethyl sulfoxide (DMSO; 0.1% (w/w) final concentration) was added to induce an enzyme reaction. After incubation at room temperature for 1 h, the fluorescence intensity (FI) of fluorescent hydrolysate of Boc-Gln-Ala-Arg-MCA (7-amino-4-methylcoumarin) were read using SpectraMax M5 plate reader (Molecular Devices, San Jose, CA, USA) with excitation of 380 nm and emission of 460 nm. The inhibitory rate of ingredients was calculated as follows: Inhibition (%) = (FI of control–FI of treatment) / FI of control × 100. The dose–response relationship between inhibition (%) and test ingredient concentration was plotted and used to determine the half maximal inhibitory concentration (IC50) using the DRC package in R software program (v3.6.1), as described in [11]. In brief, the dose–response data were fitted using a four-parametric log-logistic model and used estimate the IC50.

## Preparation of 3D structures of the target proteins for docking simulations

We prepared a human ACE2 structure that is suitable for docking simulations using a crystal structure (PDB ID: 1R4L [14]; resolution 3.00 Å). Structural refinement was performed using Homology Modeling Professional for HyperChem (HMHC) software [15, 16]. Conversely, the X-ray crystal structure of human TMPRSS2 had not been resolved at the time of submission of our manuscript to bioRixv. According to the reference [17], a human TMPRSS2 model that is suitable for docking simulations was prepared by the SWISS-MODEL [18], using a template (PDB ID: 5CE1; human serine protease complexed with inhibitor). The inhibitor of 5CE1 was extracted for the prepared model using HMHC. As the X-ray crystal structure of human TMPRSS2 (PDB ID: 7MEQ; resolution 1.95 Å; Fraser et. al., April 2021) became available when our manuscript was under review, we also prepared another human TMPRSS2 structure based on this crystal structure (PDB ID: 7MEQ) using SWISS-MODEL to refine the missing residues and atoms (side-chain rotamer of the missing Gln438 in the inhibitor-binding site was determined using the energy calculations of HMHC). Then, the N- and C-terminals of both structures were treated as zwitterions; aspartic and glutamic acid residues were treated as anions; and lysine, arginine, and histidine residues were treated as cations under the physiological conditions. Other small molecules except for the inhibitor, $Zn^{2+}$ ion, $Cl^-$ ion, and crystal waters were removed. The detailed preparation procedure was described previously in detail [19].

A 3D structure of the seven test compounds was downloaded from the PubChem website in an SDF file format. These structures were converted into individual PDBQT files under the physiological condition of pH = 7.4 using a Docking Study with HyperChem (DSHC) software [15, 16].

## Validation of the docking studies with crystal inhibitors and subsequent docking simulations for the test ingredients

The above prepared target protein structures and inhibitor structures were converted to the PDBQT files using DSHC software. Configuration files for performing AutoDock Vina [20] docking simulations were prepared based on the coordinates of the corresponding crystal inhibitor using the AutoDock Vina In Silico Screening Interface of DSHC. The exhaustiveness value was set to 100, and the top nine docking modes were maximally output. This docking condition was also used for subsequent docking simulations of the individual test compounds [19].

## Results

### Inhibitory effects of toothpaste and mouthwash ingredients on the interaction between the spike protein of SARS-CoV-2 and ACE2

To understand the effect of commercially available toothpaste and mouthwash on the SARS-CoV-2 entry mechanisms into host cells, we evaluated 30 general ingredients (Table 1). We first screened the 30 ingredients for inhibitory effects on the interaction between the spike protein of SARS-CoV-2 and ACE2 using enzyme-linked immunosorbent assay. When analyzed at 1% (w/w), 18 ingredients showed a >0% inhibitory rate (Fig 1). Five ingredients (sodium tetradecene sulfonate [TDS], sodium N-lauroyl-N-methyltaurate [LMT], sodium N-lauroylsarcosinate [LSS], copper gluconate [GCU], and sodium dodecyl sulfate [SDS]) exhibited greater than 50% inhibitory rates. The inhibitory rates of TDS, LMT, LSS, GCU, and SDS at 1% (w/w) were 96.7%, 95.9%, 94.9%, 94.5%, and 91.3%, respectively. To clarify the details of the inhibitory effects and to determine the IC50 values, we assessed the dose-effect relationship of these five ingredients. TDS, LMT, and LSS ranged from 0.00005% to 1% (w/w); SDS ranged from 0.0005% to 1% (w/w); and GCU ranged from 0.005% to 1% (w/w) (Fig 2A–2E). TDS, LMT, LSS, GCU, and SDS showed inhibitory activities against the interaction between the RBD of the spike protein and ACE2 with IC50 values of 0.009, 0.022, 0.043, 0.097, and 0.005% (w/w), respectively (Fig 2A–2E).

### Inhibitory effects of toothpaste and mouthwash ingredients on TMPRSS2 protease activity

We next screened the 30 ingredients (Table 1) for inhibitory effects on the serine protease activity of TMPRSS2. The TMPRSS2 activity test was carried out using recombinant human TMPRSS2 and a synthetic peptide substrate [12, 13]. When analyzed at 1% (w/w), 20 ingredients showed a > 0% inhibitory rate. Cetylpyridinium chloride (CPC) and sodium saccharin (SAC) clearly exhibited false-positive reactions in the assay in which the substrate, Boc-Gln-

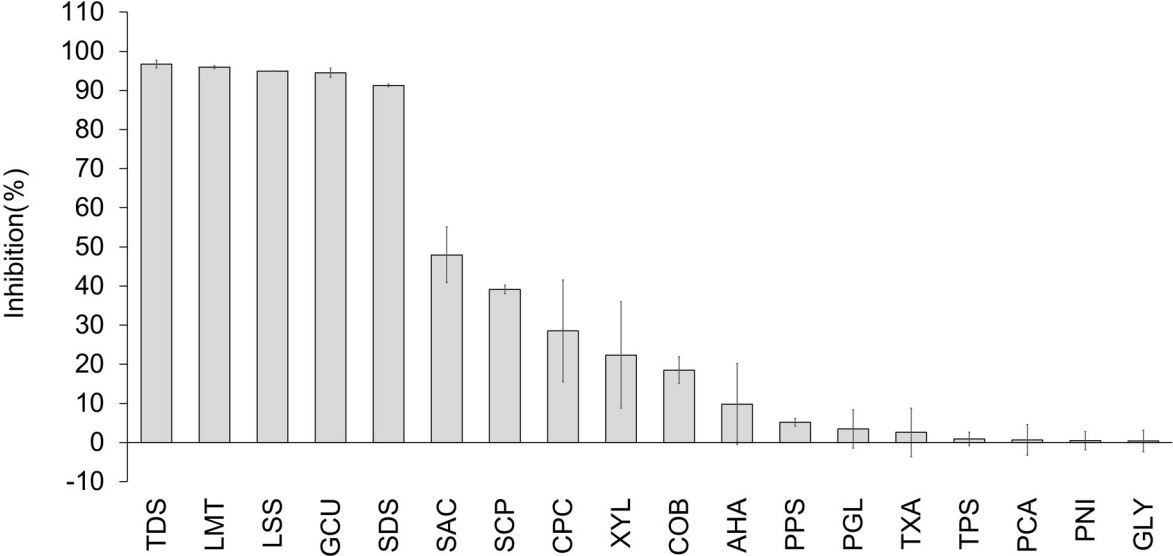

**Fig 1. Screening of toothpaste and mouthwash ingredients exhibited inhibitory effects on the interaction between the spike protein RBD of SARS-CoV-2 and ACE2.** Eighteen toothpaste and mouthwash ingredients at 1% (w/w) exhibited inhibitory effects interaction between the spike protein of SARS-CoV-2 and ACE2 *in vitro*. The interaction between the spike protein RBD of SARS-CoV-2 and ACE2 was evaluated by measuring AV at 450 nm using a microplate reader. Data are expressed as the mean ± SD (n = 3).

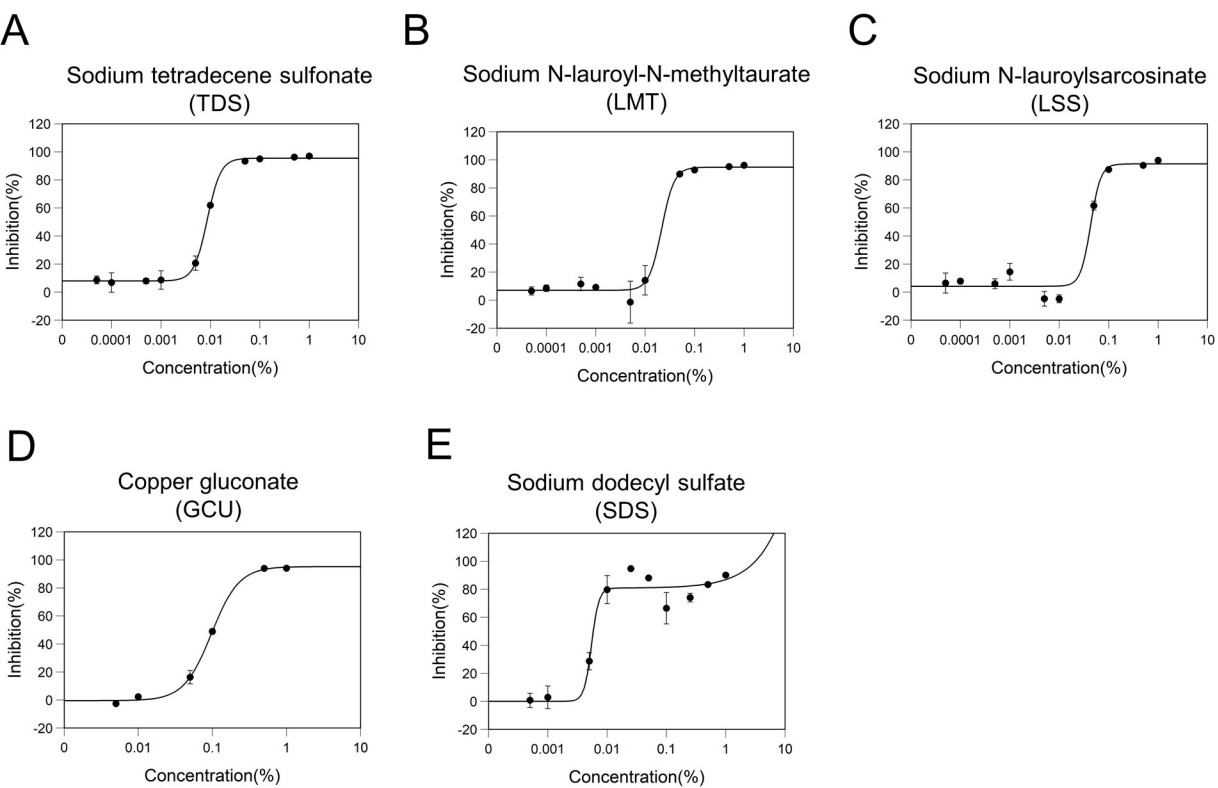

**Fig 2. Effect of ingredient concentration on inhibitory effects of the interaction between the spike protein of SARS-CoV-2 and ACE2.** The dose-response inhibitory effects of interaction between spike protein of SARS-CoV-2 and ACE2 for (A) Sodium tetradecene sulfonate, (B) Sodium N-lauroyl-N-methyltaurate, (C) Sodium N-lauroylsarcosinate, (D) Copper gluconate, and (E) Sodium dodecyl sulfate are shown. The data were plotted and modeled by a four-parameter Log-logistic fit (A–D) and a four-parameter Brain–Cousens fit (E) to determine the 50% inhibitory concentration (IC50) value. All data points were expressed as the mean ± SD (n = 3).

Ala-Arg-MCA, was replaced with 7-amino-4-methylcoumarin; therefore, CPC and SAC were excluded from this evaluation (S1 Table). Therefore, in this study, 18 ingredients with an inhibition rate > 0% (Fig 3) were selected for this study. Seven ingredients (SDS, TDS, GCU, tranexamic acid [TXA], LMT, LSS, and 6-aminohexanoic acid [AHA]) exhibited inhibitory rates above 50%. The inhibitory rates of SDS, TDS, GCU, TXA, LMT, LSS, and AHA at 1% (w/w) were 99.9%, 97.5%, 97.2%, 92.0%, 88.6%, 87.6%, and 71.7%, respectively. To clarify the details of the inhibitory effects and determine the IC50 values, we assessed the dose-effect relationship of these seven ingredients using IC50 values from 0.0005% to 1% (w/w) (Fig 4A–4G). SDS, TDS, GCU, TXA, LMT, LSS, and AHA showed inhibitory activities against the protease activity of TMPRSS2 with IC50 values of 0.014, 0.018, 0.411, 0.054, 0.098, 0.102, and 0.449% (w/w), respectively (Fig 4A–4G).

### Docking simulations of toothpaste, mouthwash, and their ingredients in the human ACE2 and human TMPRSS2 model

To support the *in vitro* assay, we performed a molecular docking simulation. In the molecular docking study, we used an X-ray crystal structure (PDB ID: 1R4L) for human ACE2 in closed conformation with a synthetic inhibitor ((*S,S*)-2-{1-carboxy-2-[3-(3,5-dichloro-benzyl)-3*H*-imidazol-4-yl]-ethylamino}-4-methyl-pentanoic acid) [14]. The closed conformation of ACE2 prevented binding to the SARS-CoV-2 spike protein [21]. To determine the docking conditions, we performed a re-docking study using the synthetic inhibitor complexed in the crystal

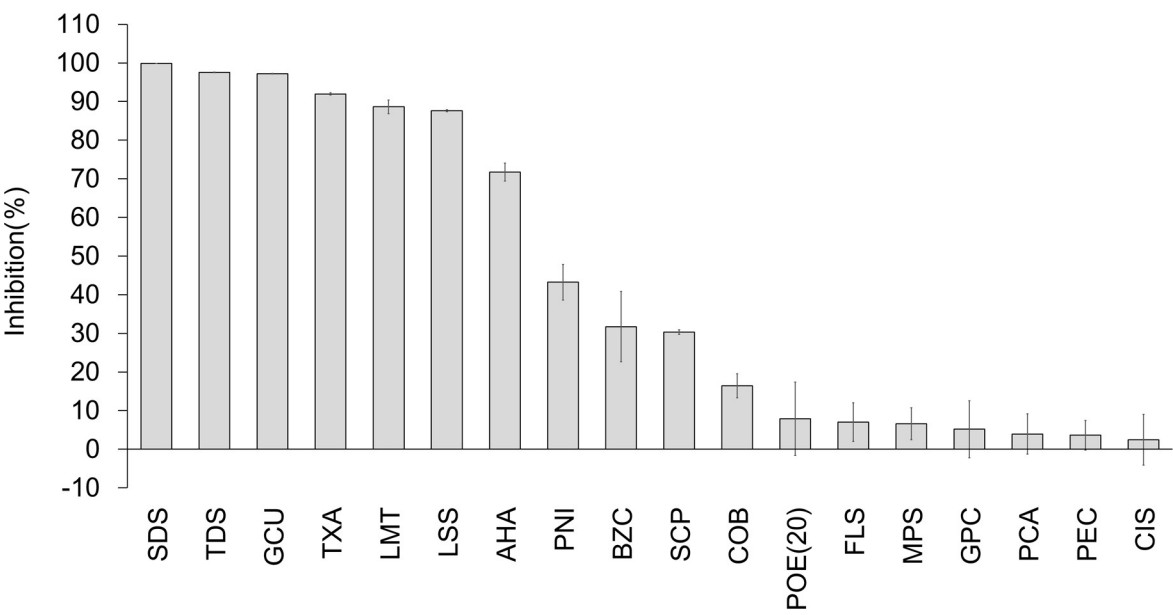

**Fig 3. Screening of toothpaste and mouthwash ingredients exhibited inhibitory effects on the serine protease activity of TMPRSS2.**
Eighteen toothpaste and mouthwash ingredients at 1% (w/w) exhibited inhibitory effects on the TMPRSS2 serine protease activity.
TMPRSS2 cleaved Boc-Gln-Ala-Arg-MCA as the substrate and produced the potent fluorophore, AMC (7-amino-4-methylcoumarin).
Values were normalized against the intensity of the absence of test ingredients. Data are expressed as the mean ± SD (n = 3).

structure. As for the results, the most stable docking mode obtained from the re-docking study reproduced the crystal structure shown in Fig 5A and 5B. The X-ray crystal structure of human TMPRSS2 had not been resolved at the time of submission of our manuscript to bioRixv. We performed docking simulations using the homology model obtained from SWISS-MODEL using a template (PDB ID: 5CE1; human serine protease complexed with an inhibitor, 2-[6-(1-hydroxycyclohexyl)pyridin-2-yl]-1*H*-indole-5-carboximidamide) as previously described [17]. Although the previous study [17] searched the binding site of the TMPRSS2 inhibitor camostat mesylate in the model using Molecular Operating Environment (MOE) software, we performed docking simulations at the crystal inhibitor-binding site. The re-docking study reproduced a crystal structure of the inhibitor, as shown in Fig 5C and 5D. While our manuscript was under review, the X-ray crystal structure of human TMPRSS2 with nafamostat became available. Upon assessment of our TMPRSS2 model against the crystal structure, we found that our TMPRSS2 model was very similar to the crystal structure (root-mean-square deviation was 0.665Å for Ca atoms in the whole structure) and that the nafamostat binding site (fragment structure of nafamostat was covalently bound to Ser441 of the inhibitor-binding site) was identical to the inhibitor-binding site of the model. All side-chain and main-chain conformations in the inhibitor-binding site of our model showed excellent agreement with those of the human TMPRSS2 crystal structure, as shown in S1 Fig. This strongly supported the results of the docking simulations using the predicted TMPRSS2 model. We also performed the docking simulations using the refined human TMPRSS2 crystal structure (PDB ID: 7MEQ).

Based on the conditions confirmed above, the docking simulations of the seven test ingredients (Figs 2 and 4, Table 2), whose IC$_{50}$ values were estimated using *in vitro* assay, were performed using AutoDock Vina program by targeting the proteins. Incidentally, ingredients used for docking simulations removed counter ions from the test ingredients (Table 2). Fig 6 shows the most stable docking mode, except for 6-aminohexanoic acid, tranexamic acid, and

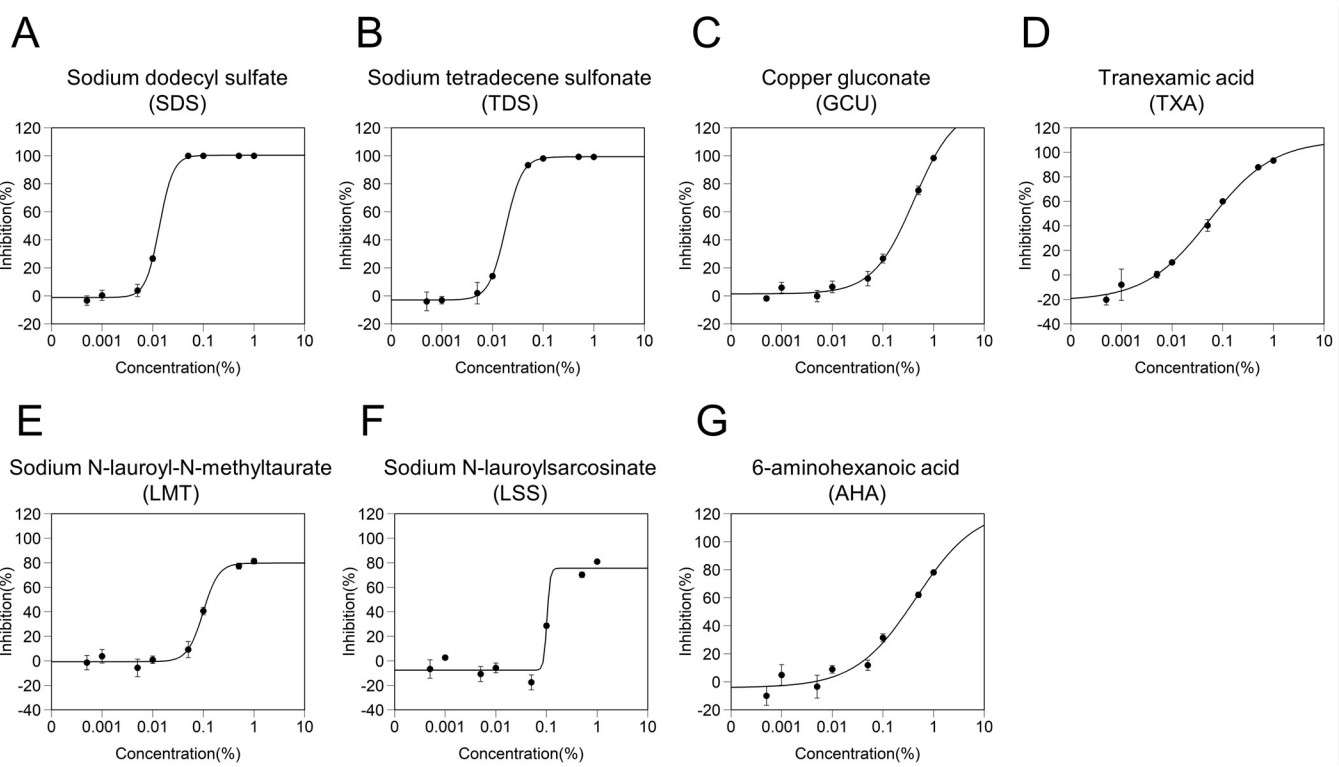

**Fig 4. Effect of ingredient concentration on inhibitory effects of serine protease activity of TMPRSS2.** The dose-response inhibitory effects of TMPRSS2 serine protease activity by (A) Sodium dodecyl sulfate, (B) Sodium tetradecene sulfonate, (C) Copper gluconate, (D) Tranexamic acid, (E) Sodium N-lauroyl-N-methyltaurate, (F) Sodium N-lauroylsarcosinate, and (G) 6-aminohexanoic acid are shown. The data were plotted and modeled by a four-parameter Log-logistic fit to determine the 50% inhibitory concentration (IC50) value. All data points expressed as the mean ± SD (n = 3).

gluconic acid used for docking with the human TMPRSS2 model, which were obtained from AutoDock Vina docking simulations. For the human TMPRSS2 model, reasonable docking modes of 6-aminohexanoic acid, tranexamic acid, and gluconic acid were observed within the top nine docking mode scores. Tables 3 and 4 show the AutoDock Vina scores (empirical binding free energy: $\Delta G_{\mathrm{bind}}$ (kcal/mol)) of these ingredients. Considering that the AutoDock Vina score is an empirical binding free energy, we expected that a score of −6 kcal/mol would theoretically present the single-digit μM binding affinity with both target proteins. The obtained AutoDock Vina scores and docking modes indicated that *N*-lauroyl-*N*-methyltaurine, *N*-lauroylsarcosine, gluconic acid, dodecyl sulfate, and (*E*)-tetradec-1-ene-1-sulfonic acid could weakly bind to human ACE2 at the inhibitor-binding site (Table 3, Fig 6A) [22, 23]. Additionally, gluconic acid and tranexamic acid could weakly bind to human TMPRSS2 at the inhibitor-binding site (Table 4, Fig 6B) [23, 24]. The docking simulations using the refined human TMPRSS2 crystal structure (PDB ID: 7MEQ) gave a similar AutoDock Vina score and docking mode for these test ingredients (S2 Fig and S2 Table). The docking modes of these test ingredients were similar for the respective target proteins (Fig 6A and 6B, and S2 Fig). These results suggest that these ingredients may function through the same underlying mechanism. These results supported the results of the *in vitro* assay.

## Discussion

In this study, we found that some general ingredients contained in commercially available toothpaste and mouthwash exhibit inhibitory effects on the SARS-CoV-2 spike protein-ACE2

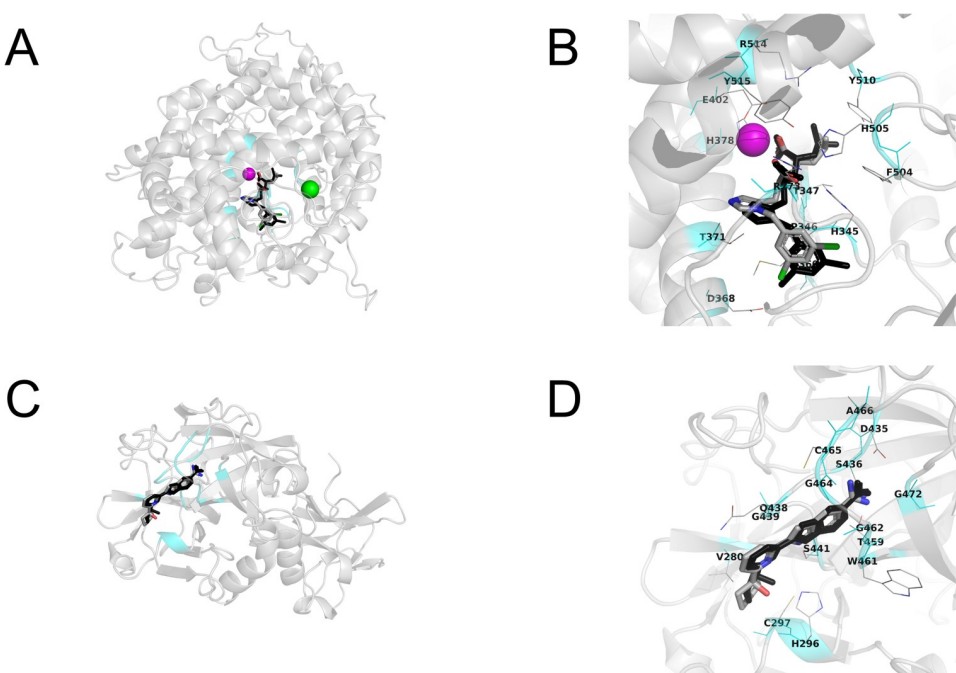

**Fig 5. Re-docking studies for human ACE2 and human TMPRSS2 model with their crystal inhibitors.** (A, B) Human ACE2. (C, D) Human TMPRSS2 model. The most stable docking mode of crystal inhibitor shows in CPK color using tubes. The original crystal structure of the inhibitor is represented by black tubes. A magenta sphere shows $Zn^{2+}$ ion and a green sphere shows the $Cl^-$ ion. The inhibitor-binding site located at 3 Å from all heavy atoms of the crystal inhibitor is shown in cyan color. Amino acid residues located at 3 Å from the inhibitor are shown using thin tubes with label. Hydrogen atoms are neglected.

interaction and TMPRSS2 serine protease activity, which are key factors in the *in vitro* SARS-- CoV-2 infection. To our knowledge, this is the first demonstration that toothpaste and mouthwash ingredients exhibit inhibitory effects on key host factors of SARS-CoV-2 entry into host cells.

In commercially available toothpastes and mouthwashes, surfactants, GCU, TXA, and AHA are typically contained at concentrations up to 2.5% [25], 0.1% [26], 0.05% [27], and 0.2% [28], respectively. Toothpastes are generally diluted by 3–4 fold by saliva during tooth brushing, and this dilution rate is commonly used in the *in vitro* studies to evaluate the effects of toothpastes containing antibacterial ingredients and/or fluoride [29–31]. When calculated based on the abovementioned concentrations and dilution rates, the estimated concentrations of each ingredient in the oral cavity are assumed to be 0.625%, 0.025%, 0.0125%, and 0.05% for

**Table 2. Test ingredients subjected to docking simulations.**

| Ingredients name | Abbreviation | Ingredients used for docking simulations | |
|---|---|---|---|
| | | Name | Pubchem CID |
| 6-aminohexanoic acid | AHA | 6-aminocaproic acid | 564 |
| Tranexamic acid | TXA | Tranexamic acid | 5526 |
| Sodium N-lauroylsarcosinate | LSS | N-Lauroylsarcosine | 7348 |
| Sodium dodecyl sulfate | SDS | Dodecyl sulfate | 8778 |
| Copper gluconate | GCU | Gluconic acid | 10690 |
| Sodium N-lauroyl-N-methyltaurine | LMT | N-Lauroyl-N-methyltaurine | 61353 |
| Sodium tetradecene sulfonate | TDS | (E)-Tetradec-1-ene-1-sulfonic acid | 6437821 |

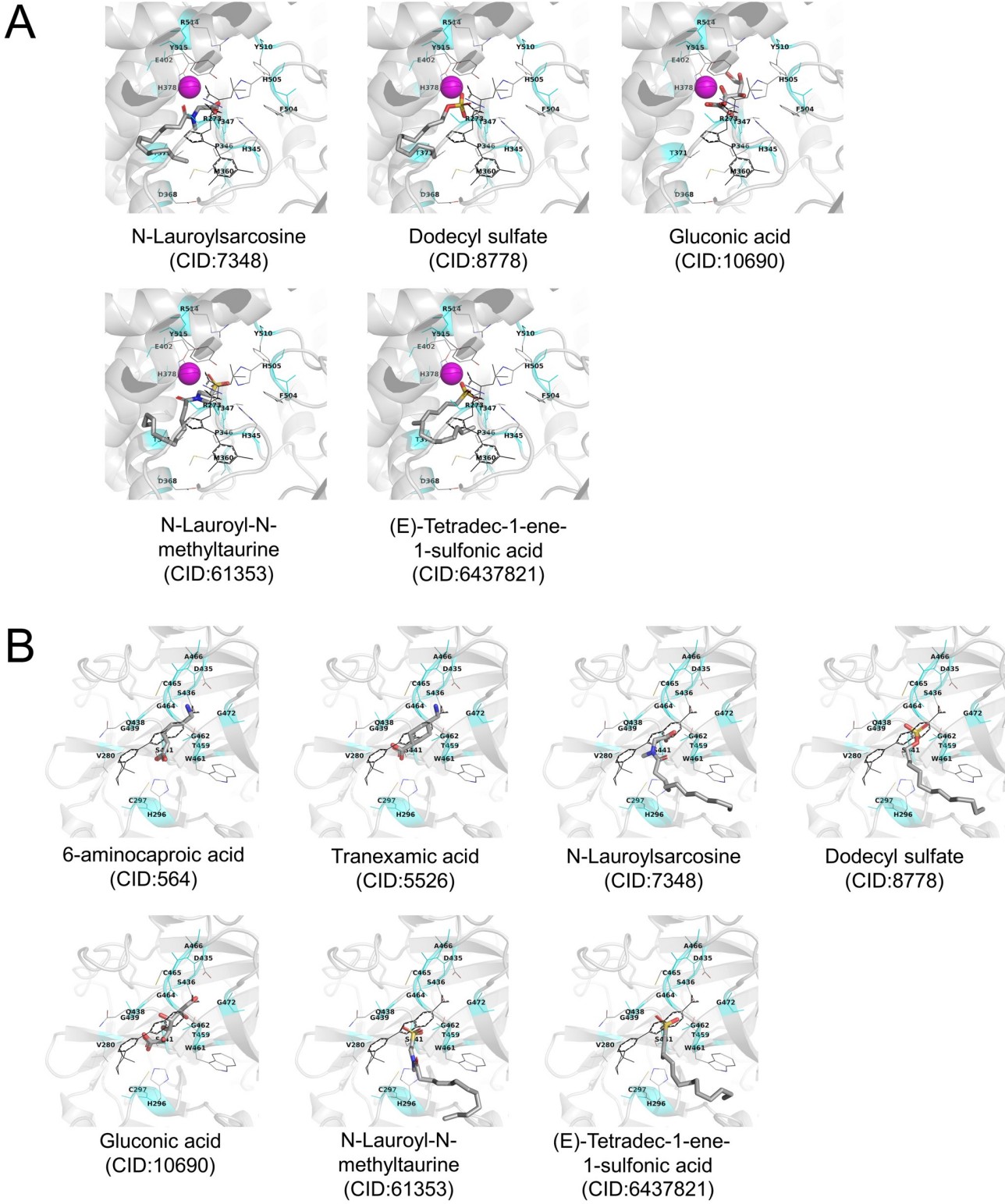

**Fig 6. Stable docking mode of the selected ingredients obtained from AutoDock Vina docking simulations.** (A) Human ACE2. (B) Human TMPRSS2 model. Stable docking mode of test compounds is shown as CPK-colored tubes. The original crystal structure of the inhibitor is shown in black lines. A magenta sphere shows $Zn^{2+}$ ion. Inhibitor-binding site located at 3 Å from all heavy atoms of the crystal inhibitor is shown in cyan color. Amino acid residues located at 3 Å from the inhibitor are shown using thin tubes with label. Hydrogen atoms are neglected.

**Table 3. Vina score of test ingredients for human ACE2.**

| Ingredients | | Vina Score (kcal/mol) |
|---|---|---|
| Name | Pubchem CID | |
| Inhibitor | - | −9.1 |
| N-Lauroyl-N-methyltaurine | 61353 | −6.6 |
| N-Lauroylsarcosine | 7348 | −6.3 |
| Gluconic acid | 10690 | −6.1 |
| Dodecyl sulfate | 8778 | −5.9 |
| (E)-Tetradec-1-ene-1-sulfonic acid | 6437821 | −5.9 |

surfactants, GCU, TXA, and AHA, respectively, when using a toothpaste (maximum combination concentration ÷ 4). The $IC_{50}$ values of the surfactant, GCU, TXA, and AHA were 0.005%–0.043% and 0.014%–0.102% (spike protein-ACE2 binding and TMPRSS2 inhibition), 0.097% and 0.411% (spike protein-ACE2 binding inhibition and TMPRSS2 inhibition), 0.054% (TMPRSS2 inhibition), and 0.449% (TMPRSS2 inhibition), respectively. In comparison with the estimated concentration of each ingredient in the oral cavity while using toothpaste, the assumable concentration of surfactants in the oral cavity was higher than the $IC_{50}$ value of both the spike protein-ACE2 binding and TMPRSS2 inhibitions, whereas the GCU, TXA, and AHA concentrations in the oral cavity were lower than the $IC_{50}$ value of the TMPRSS2 inhibition. Based on the abovementioned results, among the seven ingredients that showed spike protein-ACE2 interaction and TMPRSS2 serine protease activity inhibitory effects *in vitro*, four surfactants are presumed to be potentially present at concentrations higher than the $IC_{50}$, even when used in toothpastes. However, TXA, AHA, and GCU do not reach the expected oral concentration of $IC_{50}$ when using commercially available toothpaste, although they are expected to be effective when used at high concentrations. Furthermore, after tooth brushing using a toothpaste, rinsing with 5–20 mL water has been reported in several clinical trials [32, 33]. Therefore, the concentration of these surfactants are considered to be approximately the same or exceed the $IC_{50}$ value, even if they are assumed to get further diluted ten times by rinsing with 10 mL of water after brushing (approximate value: 0.0625%). In addition, the inhibitory effects of these ingredients (SDS, TDS, LMT, LSS, GCU, TXA, and AHA) on spike protein–ACE2 interaction and TMPRSS2 serine protease activity in the presence of saliva tended to be comparable to those in the absence of saliva (S3 and S4 Figs). These results suggest that these ingredients may have an inhibitory effect on the spike protein–ACE2 interaction and TMPRSS2 serine protease activity even in the oral cavity.

**Table 4. Vina score of test ingredients for human TMPRSS2 model.**

| Ingredients | | Vina Score (kcal/mol) |
|---|---|---|
| Name | Pubchem CID | |
| Inhibitor | - | −8.4 |
| Gluconic acid | 10690 | −5.8 |
| Tranexamic acid | 5526 | −5.5 |
| N-Lauroylsarcosine | 7348 | −5.4 |
| Dodecyl sulfate | 8778 | −5.2 |
| N-Lauroyl-N-methyltaurine | 61353 | −5.1 |
| (E)-Tetradec-1-ene-1-sulfonic acid | 6437821 | −5.0 |
| 6-aminocaproic acid | 564 | −4.0 |

In general, anionic surfactants are known to exhibit a protein denaturing effect [34–36], and we considered that the inhibitory effect of the surfactants in this study was exerted by the denaturation of ACE2 or that of the spike protein and TMPRSS2. Previous studies demonstrated that the RBD of the SARS-CoV-2 spike protein was positively charged at physiological pH values (the theoretical isoelectric point of RBD was 8.9) [37, 38]. At the PBS pH values of 7.0–7.3 used in this study, the spike protein was presumed to be positively charged. Therefore, anionic surfactants are more likely to bind to the RBD surface of the SARS-CoV-2 spike protein than cationic or nonionic surfactants. As a result, the denaturation of ACE2 and spike proteins is induced, which may disrupt the conformation of both proteins and consequently exert an inhibitory effect. In addition, Neufurth et al. suggested that anionic inorganic polyphosphate (poly-P) binds to the RBD surface and prevents its binding to ACE2 [38]. In the oral cavity, pH is maintained at near neutrality (6.7–7.3) by the saliva [39]; thus, RBD of the SARS-CoV-2 spike protein might be positively charged in the oral cavity. Therefore, these anionic surfactants and spike protein RBD interactions might also result in the inhibitory effect. However, because the isoelectric point of ACE2 is 5.6, it is unlikely that a positively charged reaction similar to that of the spike protein has occurred. On the other hand, a surfactant (SDS) could reportedly dock to a protein as a single molecule, resulting in protein function changes [40]. Hence, we focused on the specific binding of the spike protein RBD and ACE2. The spike protein RBD is not identified as an obvious binding site for inhibitors affecting the spike protein binding with ACE2, although ACE2 was identified as a binding site for inhibitors; thus, we performed a molecular docking simulation for human ACE2. The results showed that these four surfactants could bind to the inhibitor-binding site of human ACE2, seeming to be a part of the mechanism in addition to denaturation.

Additionally, since the theoretical isoelectric point of TMPRSS2 is 8.58 [41], TMPRSS2 is presumably positively charged at the pH of the assay buffer used in this study. Therefore, anionic surfactants are more likely to bind to TMPRSS2 than cationic or nonionic surfactants, thereby leading to the induction of protein denaturation and the disruption of TMPRSS2 conformation, resulting in an inhibitory effect. As a near-to-neutral (pH 6.7–7.3) pH is maintained in the oral cavity [39], the same electrostatic interaction would also occur there.

The GCU also exhibited inhibitory effects on the interaction between the spike protein and ACE2, as well as the TMPRSS2 serine protease activity. Docking simulations predicted that gluconic acid, which is one of parent components of GCU, could bind to both inhibitor-binding sites of human ACE2 and TMPRSS2 (Tables 3 and 4). These results suggested that gluconic acid could inhibit spike protein-ACE2 interaction and TMPRSS2 serine protease activity. However, whether gluconic acid or copper ions exhibit an actual inhibitory effect is unknown. An assay for gluconic acid or copper ions alone could elucidate the mechanism of the inhibitory effect by GCU.

TXA and AHA exhibit inhibitory effects on serine protease plasminogen [42]. In the *in vitro* assay, TXA and AHA exhibited inhibitory effects on TMPRSS2 protease activity (Fig 4). However, docking simulations predicted that TXA, but not AHA, could bind to the inhibitor-binding sites of TMPRSS2 (Table 4). Therefore, TXA possibly inhibited the TMPRSS2 protease activity by binding to TMPRSS2. However, it is unclear how AHA inhibited the TMPRSS2 protease activity in this study. Rahman et al. suggested that TMPRSS2 inhibitor camostat mesylate can bind to another inhibitor-binding site of TMPRSS2, whose site was estimated by MOE [17]. There exists a possibility of AHA binding to another or unidentified inhibitor-binding site in TMPRSS2.

Currently, various SARS-CoV-2 variants have emerged and are attracting attention as a cause of the spread of the infection. The major mutants [Alpha (B.1.1.7), Gamma (P.1), and Delta (B.1.617.2)], as of August 2021, have characteristic substitutions such as E484K, N501Y,

D614G, or L452R in the spike protein. These substitutions do not change the binding site of ACE2 to the spike protein but rather increase the affinity between the spike protein and ACE2 [43–47]. Unlike neutralizing antibodies against SARS-CoV-2 spike protein, ingredients found in this study (SDS, TDS, LMT, LSS, and GCU) have inhibitory effects on ACE2, so it is considered that they have inhibitory effects on these SARS-CoV-2 variants as well. In addition, the ingredients found in this study (SDS, TDS, LMT, LSS, GCU, and TXA) have been shown to inhibit the serine protease activity of TMPRSS2, which is involved in the viral infection of cells after ACE2 and spike protein binding (Figs 3, 4 and 6 and Table 4). Therefore, even if the spike protein in these SARS-CoV-2 variants binds to ACE2, these ingredients (SDS, TDS, LMT, LSS, GCU, and TXA) will inhibit TMPRSS2-dependent cell membrane fusion and thereby prevent infection of these SARS-CoV-2 variants to host cells. Considering this information, the selected toothpaste and mouthwash ingredients in this study (SDS, TDS, LMT, LSS, GCU, and TXA) may have an inhibitory effect on the infection with SARS-CoV-2 variants in the oral cavity.

This study's findings must be seen in the light of some limitations. The inhibitory effects of the six ingredients on ACE2 and TMPRSS2 are based on an *in vitro* molecular study and *in silico* analysis and not on actual *in vivo* studies; it remains unclear whether SARS-CoV-2 infection could be suppressed *in vivo*. However, it is potentially important that these toothpaste and mouthwash ingredients can potentially prevent SARS-CoV-2 infection. Therefore, we propose an anti-SARS-CoV-2 experiment using oral mucosa and upper respiratory tract cells and a virus infectivity evaluation test using epithelial culture cells overexpressing human ACE2 and TMPRRSS2 and pseudovirus expressed SARS-CoV-2 spike protein. Furthermore, clinical trials using these ingredients or toothpastes and mouthwashes containing these ingredients would help in further elucidating the antiviral activities of these ingredients and oral care products containing these ingredients against SARS-CoV-2 infection.

## Conclusions

We found that six general ingredients contained in commercially available toothpaste and mouthwash exhibited inhibitory effects on the interaction between the RBD of the spike protein and ACE2 and the TMPRSS2 protease activity. As ACE2 and TMPRSS2 are vital for the entry of SARS-CoV-2 into host cells, the five ingredients (SDS, TDS, LMT, LSS, and GCU) that were effective against ACE2 and TMPRSS2 may exert inhibitory effects in two steps, and a highly preventive effect on SARS-CoV-2 infection can be expected. Additionally, TXA exerted inhibitory effects on TMPRSS2 protease activity. Therefore, our findings suggest that these toothpaste and mouthwash ingredients could help prevent SARS-CoV-2 infection. However, further experimental and clinical studies are necessary to elucidate these mechanisms.

## Supporting information

**S1 File. Supplemental methods.**
(DOCX)

**S1 Fig. Superposition of humanTMPRSS2 homology model prepared from 5CE1 template (shown in red color) on the human TMPRSS2 crystal structure (PDB ID: 7MEQ; shown in blue color).** Inhibitor-binding site (the residues located at 5Å from the inhibitor of 5CE1 (shown in red tubes)) was shown in this figure. The Gln438 side-chain atoms of crystal structure (blue) were not observed in 7MEQ.
(TIF)

**S2 Fig. Stable docking mode of the selected ingredients obtained from AutoDock Vina docking simulations using the refined human TMPRSS2 crystal structure (PDB ID: 7MEQ).** The stable docking mode of test compounds is shown as CPK-colored tubes. The inhibitor-binding site located at 3 Å from all heavy atoms of the crystal inhibitor is shown in cyan color. Amino acid residues located at 3 Å from the inhibitor are shown using thin tubes with labels. Hydrogen atoms are neglected.
(TIF)

**S3 Fig. Effect of ingredient on inhibitory effects in solutions containing saliva of the SARS-CoV-2 spike protein-ACE2 interaction.** The inhibitory effects in assay solutions with saliva (10% (v/v)) or without saliva (0%(v/v)) of SARS-CoV-2 spike protein-ACE2 interaction by (A) Sodium dodecyl sulfate, (B) Sodium N-lauroyl-N-methyltaurate, (C) Sodium tetradecene sulfonate, (D) Sodium N-lauroylsarcosinate, and (E) Copper gluconate are shown. For the data in the assay solution without saliva, some of the data in Fig 2 are republished to compare the effect with the data in the assay solution with saliva. All data points were expressed as the mean ± SD (n = 3).
(TIF)

**S4 Fig. Effect of ingredient on inhibitory effects in solutions containing saliva of the serine protease activity of TMPRSS2.** The inhibitory effects in assay solutions with saliva (10% (v/v)) or without saliva (0%(v/v)) of TMPRSS2 serine protease activity by ((A) Sodium dodecyl sulfate, (B) Sodium N-lauroyl-N-methyltaurate, (C) Sodium tetradecene sulfonate, (D) Sodium N-lauroylsarcosinate, and (E) Copper gluconate, (F) Tranexamic acid, and (G) 6-aminohexanoic acid are shown. For the data in the assay solution without saliva, some of the data in Fig 4 are republished to compare the effect with the data in the assay solution with saliva. All data points were expressed as the mean ± SD (n = 3).
(TIF)

**S1 Table. Effect of ingredient on fluorescent substance (7-Amino-4-methylcoumarin).**
(DOCX)

**S2 Table. Vina score of test ingredients for the refined human TMPRSS2 crystal structure (7MEQ).**
(DOCX)

**S1 Data. Raw data of the data shown in Figs 1–4, Tables 3 and 4, S3 and S4 Figs, and S1 and S2 Tables.**
(XLSX)

## Acknowledgments

The authors thank Dr. Yasushi Kakizawa of the Lion Corporation for his cooperation in conducting this study. The authors would also like to thank Enago (https://www.enago.jp) for the English language review.

## Author Contributions

**Conceptualization:** Taku Iwamoto, Kota Tsutsumi, Satoru Morishita, Kei Kurita, Yukio Yamamoto, Eiji Nishinaga, Keiichi Tsukinoki.

**Formal analysis:** Riho Tateyama-Makino, Mari Abe-Yutori.

**Funding acquisition:** Yukio Yamamoto, Eiji Nishinaga.

**Investigation:** Riho Tateyama-Makino, Mari Abe-Yutori, Motonori Tsuji.

**Methodology:** Riho Tateyama-Makino, Mari Abe-Yutori, Taku Iwamoto, Kota Tsutsumi, Motonori Tsuji.

**Project administration:** Yukio Yamamoto, Eiji Nishinaga.

**Software:** Motonori Tsuji.

**Supervision:** Keiichi Tsukinoki.

**Validation:** Riho Tateyama-Makino, Mari Abe-Yutori, Motonori Tsuji.

**Visualization:** Riho Tateyama-Makino, Mari Abe-Yutori, Taku Iwamoto, Kota Tsutsumi, Motonori Tsuji.

**Writing – original draft:** Riho Tateyama-Makino, Mari Abe-Yutori, Taku Iwamoto, Kota Tsutsumi, Motonori Tsuji.

**Writing – review & editing:** Satoru Morishita, Kei Kurita, Yukio Yamamoto, Eiji Nishinaga, Keiichi Tsukinoki.

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
