## [Decision Letter · Decision Letter 0]

29 Jul 2021

PONE-D-21-20049

The inhibitory effects of toothpaste and mouthwash ingredients on the interaction between the SARS-CoV-2 spike protein and ACE2, and the protease activity of TMPRSS2 in vitro

PLOS ONE

Dear Dr. Tateyama-Makino,

Thank you for submitting your manuscript to PLOS ONE. After careful consideration, we feel that it has merit but does not fully meet PLOS ONE’s publication criteria as it currently stands. Therefore, we invite you to submit a revised version of the manuscript that addresses the points raised during the review process.

Please attention to the methods as the reviewer commented.

We look forward to receiving your revised manuscript.

Kind regards,

Etsuro Ito

Academic Editor

PLOS ONE

Journal Requirements:

2. Thank you for providing the following Funding Statement: 

“This work was fully funded by Lion Corporation (https://www.lion.co.jp/en/). R Tateyama-Makino, M Abe-Yutori, T Iwamoto, K Tsutsumi, S Morishita, K Kurita, Y Yamamoto, and E Nishinaga are employees of Lion Corporation. The Lion Corporation provided support in the form of salaries for authors R Tateyama-Makino, M Abe-Yutori, T Iwamoto, K Tsutsumi, S Morishita, K Kurita, Y Yamamoto, and E Nishinaga . M Tsuji is a President of the Institute of Molecular Function (Saitama, Japan). Molecular docking simulation was performed by the Institute of Molecular Function under a consignment from the Lion Corporation. K Tsukinoki has received fees for technical guidance from the Lion Corporation. The funders had no role in study design, data collection and analysis, decision to publish, or preparation of the manuscript.”

We note that one or more of the authors is affiliated with the funding organization, indicating the funder may have had some role in the design, data collection, analysis or preparation of your manuscript for publication; in other words, the funder played an indirect role through the participation of the co-authors.

If the funding organization did not play a role in the study design, data collection and analysis, decision to publish, or preparation of the manuscript and only provided financial support in the form of authors' salaries and/or research materials, please review your statements relating to the author contributions, and ensure you have specifically and accurately indicated the role(s) that these authors had in your study in the Author Contributions section of the online submission form. Please make any necessary amendments directly within this section of the online submission form.  Please also update your Funding Statement to include the following statement: “The funder provided support in the form of salaries for authors [insert relevant initials], but did not have any additional role in the study design, data collection and analysis, decision to publish, or preparation of the manuscript. The specific roles of these authors are articulated in the ‘author contributions’ section.”

If the funding organization did have an additional role, please state and explain that role within your Funding Statement.

Please also provide an updated Competing Interests Statement declaring this commercial affiliation along with any other relevant declarations relating to employment, consultancy, patents, products in development, or marketed products, etc. 

Reviewers' comments:

Reviewer's Responses to Questions

**Comments to the Author**

1. Is the manuscript technically sound, and do the data support the conclusions?

Reviewer #1: Yes

Reviewer #2: No

Reviewer #3: Partly

2. Has the statistical analysis been performed appropriately and rigorously? 

Reviewer #1: N/A

Reviewer #2: No

Reviewer #3: No

3. Have the authors made all data underlying the findings in their manuscript fully available?

Reviewer #1: Yes

Reviewer #2: Yes

Reviewer #3: Yes

4. Is the manuscript presented in an intelligible fashion and written in standard English?

Reviewer #1: Yes

Reviewer #2: Yes

Reviewer #3: Yes

5. Review Comments to the Author

Reviewer #1: Dear authors,

In this report, authors described inhibitory effect of toothpaste and mouthwash thirty ingredients on the SARS-CoV-2 infection mechanism, that SARS-CoV-2 spike protein-ACE2 interaction and TMPRSS2 protease activity.　 The interaction between SARS-CoV-2 spike protein and ACE2 was calculated using the ELISA-based Inhibitor Spike S1 (SARS-CoV-2): ACE2 Screening Colorimetric Assay Kit (Figure 1 and 2). In addition, the TMPRSS2 protease activity is investigated using the release of 7-amino-4-methylcoumarin as an indicator by hydrolysis of the fluorescent substrate Boc-Gln-Ala-Arg-MCA (Figure 3 and 4). Furthermore, the interactions of the candidate ingredients with human ACE2 and human TMPRSS2 were simulated in silico (Figure 5 and 6).

The reviewer appreciates the attempt to find a candidate ingredient with guaranteed safety that is closely related to our lives against SARS-CoV-2 infections that are spreading worldwide. However, although it is mentioned in the discussion, there are some concerns about the methodology of the research, especially the evaluation of surfactant ingredients. Also, despite the in vitro and in silico assays, the assessment of interactions seems to be subjective. Despite some concerns, the reviewers found the study to provide important insights in the societal challenge of suppressing SARS-CoV-2 infections.

Remarks by reviewers

#1 Interaction between spike proteins and ACE2

The analysis of the effect of candidate components on the interaction between spike proteins and ACE2 has been investigated by ELISA-based kits. In the instructions of this kit, the candidate ingredient solution is added to the Spike S1-coated wells and incubated for 1 hour, and then the ACE2-Biotin solution is added and incubated for 1 hour. In the discussion, the concentration of the candidate ingredient solution was described, but there was no description of how long the candidate ingredient coexisted with the interacting protein. Especially in the case of surfactants, prolonged interaction with the protein is expected to affect the protein's conformation, and this is seen as due to the disruption of the conformation rather than interference of the interaction by the ingredient. If the interference of the interaction includes the disruption of the conformation, the reviewer's concern will be resolved. In this case, there is concern that the normal function of ACE2 may be interfered with. Is there any effect of surfactants on the function of ACE2 in the oral mucosa?

#2 In various parts of the text, the word "weakly" is used to describe the intensity of the interaction. If this study is an in vitro or in silico analysis, and not specifically tested by statistical analysis, it would seem that a well-founded criterion would be needed to indicate the strength of the interaction.

The evaluation method for the strength of the interaction should be described in the method section.

#3 In Figure 2 and Figure 4

The names of the candidate ingredients should be listed in the figure, as shown in Figure 6.

#4　　In table 3

The title of the table is "Vina score of test ingredients for human ACE2", but it should be "Vina score of test ingredients for human ACE2 model" as in Table 4.

Sincerely yours,

Reviewer #2: Tateyama-Makino et al. reported the actions of ingredients in toothpaste and mouthwash expected to prevent the SARS-CoV2 infection in the oral cavity. They analyzed the inhibitory effects on the interaction between the receptor-binding domain of spike protein of SARS-CoV2 and the host receptor human ACE2, as well as human TMPRSS2, which is necessary for the viral entry to the host cells. They confirmed the effect by a docking study of the ingredients showing the high inhibitory effects to the models of human ACE2 and TMPRSS2.

Since the salivary glands and mucosae in the oral cavity are considered to be significant points for SARS-CoV-2 infection and transmission, the study could have potential importance for COVID-19 prevention. However, there are several serious concerns found in the study and the conclusion is not sufficiently supported by the data, as follows.

1. Surfactants in toothpaste and mouthwash are expected to affect the envelope of the virus, and thus the action might serve as the primary effect for prevention of virus infection. However, this basic effect is not discussed in the manuscript. Therefore, the significance of the actions to ACE2 and TMPRSS2, if any, among the total effects is unclear.

2. As the authors wrote in the Discussion, the inhibitory effects of the ingredients tested in this study are most probably derived from the denaturation of the target proteins, which are induced by non-specific binding of the ingredients to the targets. Nevertheless, the authors performed the docking study, a methodology assuming specific binding, and discussed the action based on the models. However, a docking study itself does not serve as evidence for specific binding to the targets, and should be performed based on other experimental results indicating specific binding. Nevertheless, no such evidence was provided in the manuscript. In addition, the apparent reduction of the activities might also be caused by the denaturation of the probe proteins for detection in the assay kit, such as HRP.

3. Rationales of model selection for docking studies are unclear. Especially, although the authors described that a crystal structure of human TMPRSS2 has never been resolved, the structure with an inhibitor was released on Apr. 21, 2021 (PDB ID: 7MEQ).

Reviewer #3: In this manuscript, the authors found that general ingredients of toothpastes and mouthwashes possess inhibitory effects on the SARS-Cov2 spike protein-ACE2 interaction and the TMPRSS2 protease activity. Molecular docking study also revealed that the promising ingredients could bind to host factors at the inhibitor-binding site. The authors present some interesting possibilities that oral care products containing effective ingredients are able to reduce the viral load in the oral cavity of SARS-CoV-2 infected individuals. However, it is difficult to conceive that the experimental systems presented here is a physiological model. Additional experiments would be required to draw convincing conclusions in this study. The following comments are provided for the authors consideration.

Specific comments:

1. The evidence for the inhibitory effects of toothpaste and mouthwash ingredients against SARS-CoV-2 infection seems largely circumstantial, and conclusions are poorly supported, as concerned by the authors. At least, the authors should confirm that viral infectivity on the epithelial cells, such as HEK-293T overexpressing the human ACE2, Vero E6, Calu-3, is reduced in the presence of the dental care ingredients, using SARS-CoV-2 protein pseudovirus system. Addressing this comment would strengthen the conclusion.

2. Figs. 1-4: The inhibitory effects of toothpaste and mouthwash ingredients diluted in PBS were evaluated in this study, which may significantly differ from a complex environment of the human oral cavity. It would be informative to see if the saliva components, such as enzymes and serum proteins, effects on the interaction between these ingredients and ACE2 or TMPRSS2.

3. Are the ingredients effective in preventing infections with the SARS-CoV-2 variants? The authors should address this point in their discussion.

4. It is not clear from the Methods how the quantification of IC50 was carried out.

Minor points

1. Fig. 1: IBE and CPB are not listed in Table 1.

2. Fig. 6: The quality of model drawings is too low. It is difficult to evaluate.

6. PLOS authors have the option to publish the peer review history of their article (what does this mean?). If published, this will include your full peer review and any attached files.

Reviewer #1: No

Reviewer #2: No

Reviewer #3: No

---

## [Author Response · Author response to Decision Letter 0]

3 Sep 2021

Response to Journal Requirements:

RESPONSE: We have revised the formatting of the affiliations according to the abovementioned style guidelines. We have also revised the names of the Supporting Information data files from “Supporting information_makino_20210618” to “S1 data.” We hope that these revisions now adhere to the journal’s style requirements.

2. Thank you for providing the following Funding Statement: 

“This work was fully funded by Lion Corporation (https://www.lion.co.jp/en/). R Tateyama-Makino, M Abe-Yutori, T Iwamoto, K Tsutsumi, S Morishita, K Kurita, Y Yamamoto, and E Nishinaga are employees of Lion Corporation. The Lion Corporation provided support in the form of salaries for authors R Tateyama-Makino, M Abe-Yutori, T Iwamoto, K Tsutsumi, S Morishita, K Kurita, Y Yamamoto, and E Nishinaga . M Tsuji is a President of the Institute of Molecular Function (Saitama, Japan). Molecular docking simulation was performed by the Institute of Molecular Function under a consignment from the Lion Corporation. K Tsukinoki has received fees for technical guidance from the Lion Corporation. The funders had no role in study design, data collection and analysis, decision to publish, or preparation of the manuscript.”

We note that one or more of the authors is affiliated with the funding organization, indicating the funder may have had some role in the design, data collection, analysis or preparation of your manuscript for publication; in other words, the funder played an indirect role through the participation of the co-authors.

If the funding organization did not play a role in the study design, data collection and analysis, decision to publish, or preparation of the manuscript and only provided financial support in the form of authors' salaries and/or research materials, please review your statements relating to the author contributions, and ensure you have specifically and accurately indicated the role(s) that these authors had in your study in the Author Contributions section of the online submission form. Please make any necessary amendments directly within this section of the online submission form. Please also update your Funding Statement to include the following statement: “The funder provided support in the form of salaries for authors [insert relevant initials], but did not have any additional role in the study design, data collection and analysis, decision to publish, or preparation of the manuscript. The specific roles of these authors are articulated in the ‘author contributions’ section.”

If the funding organization did have an additional role, please state and explain that role within your Funding Statement.

Please also provide an updated Competing Interests Statement declaring this commercial affiliation along with any other relevant declarations relating to employment, consultancy, patents, products in development, or marketed products, etc. 

RESPONSE: The funding organization (Lion Corporation) did not play any role in the study design, data collection and analysis, decision to publish, or preparation of the manuscript. It only provided financial support in the form of authors’ salaries and/or research materials. Therefore, we stated “The funders had no role in study design, data collection and analysis, decision to publish, or preparation of the manuscript.” in the original Funding Statement. We also reviewed the Author Contributions section of the online submission form and confirm that the role(s) that the authors had in this study have been detailed accurately. Therefore, we added to the following sentence to the Funding Statement: The specific roles of these authors are articulated in the “author contributions” section.

This work was fully funded by Lion Corporation (https://www.lion.co.jp/en/). R Tateyama-Makino, M Abe-Yutori, T Iwamoto, K Tsutsumi, S Morishita, K Kurita, Y Yamamoto, and E Nishinaga are employees of Lion Corporation. Lion Corporation provided support in the form of salaries for authors R Tateyama-Makino, M Abe-Yutori, T Iwamoto, K Tsutsumi, S Morishita, K Kurita, Y Yamamoto, and E Nishinaga. M Tsuji is the President of the Institute of Molecular Function (Saitama, Japan). Molecular docking simulation was performed by the Institute of Molecular Function under a consignment from Lion Corporation. K Tsukinoki has received fees for technical guidance from Lion Corporation. The funders had no role in study design, data collection and analysis, decision to publish, or preparation of the manuscript. The specific roles of these authors are articulated in the “author contributions” section.

There are no changes in the Competing Interests Statement. Declarations related to employment and consultants have already been described in the original text. As of today (Sep 3rd, 2021), there are no published patents related to this research. This study was performed on general toothpaste and mouthwash ingredients. The ingredients evaluated in this study do not refer only to products in development at Lion Corporation or those marketed by Lion Corporation. This information will be added to the Competing Interests Statement as necessary.

RESPONSE: The phrase “data not shown” in the manuscript (P10, line 187) has been deleted, and Materials and Methods and data (S1 Table) have been added as Supporting Information. The text in the manuscript has also been revised as follows.

P10, lines 184–187: Cetylpyridinium chloride (CPC) and sodium saccharin (SAC) clearly exhibited false-positive reactions in the assay in which the substrate, Boc-Gln-Ala-Arg-MCA, was replaced with 7-amino-4-methyl coumarin; therefore, CPC and SAC were excluded from this evaluation (data not shown S1 Table).

Response to Reviewer #1:

In this report, authors described inhibitory effect of toothpaste and mouthwash thirty ingredients on the SARS-CoV-2 infection mechanism, that SARS-CoV-2 spike protein-ACE2 interaction and TMPRSS2 protease activity.　 The interaction between SARS-CoV-2 spike protein and ACE2 was calculated using the ELISA-based Inhibitor Spike S1 (SARS-CoV-2): ACE2 Screening Colorimetric Assay Kit (Figure 1 and 2). In addition, the TMPRSS2 protease activity is investigated using the release of 7-amino-4-methylcoumarin as an indicator by hydrolysis of the fluorescent substrate Boc-Gln-Ala-Arg-MCA (Figure 3 and 4). Furthermore, the interactions of the candidate ingredients with human ACE2 and human TMPRSS2 were simulated in silico (Figure 5 and 6).

The reviewer appreciates the attempt to find a candidate ingredient with guaranteed safety that is closely related to our lives against SARS-CoV-2 infections that are spreading worldwide. However, although it is mentioned in the discussion, there are some concerns about the methodology of the research, especially the evaluation of surfactant ingredients. Also, despite the in vitro and in silico assays, the assessment of interactions seems to be subjective. Despite some concerns, the reviewers found the study to provide important insights in the societal challenge of suppressing SARS-CoV-2 infections.

RESPONSE: We wish to express our appreciation to the reviewer for their insightful comments and suggestions on our paper that have helped us significantly improve it. We have addressed the reviewer’s comments as detailed below.

Remarks by the reviewer

#1 Interaction between spike proteins and ACE2 The analysis of the effect of candidate components on the interaction between spike proteins and ACE2 has been investigated by ELISA-based kits. In the instructions of this kit, the candidate ingredient solution is added to the Spike S1-coated wells and incubated for 1 hour, and then the ACE2-Biotin solution is added and incubated for 1 hour. In the discussion, the concentration of the candidate ingredient solution was described, but there was no description of how long the candidate ingredient coexisted with the interacting protein. Especially in the case of surfactants, prolonged interaction with the protein is expected to affect the protein's conformation, and this is seen as due to the disruption of the conformation rather than interference of the interaction by the ingredient. If the interference of the interaction includes the disruption of the conformation, the reviewer's concern will be resolved. In this case, there is concern that the normal function of ACE2 may be interfered with. Is there any effect of surfactants on the function of ACE2 in the oral mucosa?

RESPONSE: Thank you for your question. We agree that further details should be included in the Materials and Methods section of the manuscript to make it more comprehensible. Accordingly, we have added the following information about the coexistence time of spike protein and ACE2 with toothpaste and mouthwash ingredients:

P5-6, lines 84–88: Test ingredients were added to a 96-well plate coated with Spike S1 and incubated for 1 hour. Subsequently, ACE2-biotin solution was added to the wells, followed by incubation for 1 hour. After washing the plate, streptavidin-HRP was added and incubated for 1 hour. After washing the plate again, the HRP substrate was added, and the absorbance of the solution was measured. 

As described above, toothpaste and mouthwash ingredients coexisted with the spike protein for a total of 2 hours and with ACE2 for 1 hour. Considering the reaction time described above, the denaturing action of surfactants may disrupt the conformation of spike protein and ACE2, thereby resulting in the inhibition of spike protein–ACE2 binding. Discussion on the protein denaturation effect of surfactants was previously described on P17, lines 318–320 in the manuscript. Nevertheless, the manuscript has been revised as follows to clearly state the possibility of “the disruption of protein conformation by the surfactant.”

P17, lines 323–327: Therefore, anionic surfactants are more likely to bind to the RBD surface of the SARS-CoV-2 spike protein than cationic or nonionic surfactants. As a result, the denaturation of ACE2 and spike proteins is induced, which may disrupt the conformation of both proteins and consequently exert an inhibitory effect.

P18, lines 343–345: Therefore, anionic surfactants are more likely to bind to TMPRSS2 than cationic or nonionic surfactants, thereby leading to the induction of protein denaturation and the disruption of TMPRSS2 conformation, resulting in an inhibitory effect.

The function of ACE2 in the oral mucosa is currently unknown, except that it is possibly involved in the infection of several coronaviruses, including SARS-CoV-2. The concentrations of the surfactants evaluated in this study are within the range of concentrations generally used in toothpastes and mouthwashes and have been confirmed to pose no safety or health issues in normal use. In addition, ACE2 plays a role in lowering blood pressure by catalyzing the hydrolysis of angiotensin II into angiotensin (1-7) in the heart, lungs, and kidneys. However, there are no studies reporting that the use of toothpaste or mouthwash inhibits this hypotensive action of ACE2. Based on this, the concentration range of the surfactants used in this study would not affect the maintenance of biological functions involving ACE2 when used in the form of a toothpaste or mouthwash.

#2 In various parts of the text, the word "weakly" is used to describe the intensity of the interaction. If this study is an in vitro or in silico analysis, and not specifically tested by statistical analysis, it would seem that a well-founded criterion would be needed to indicate the strength of the interaction.

The evaluation method for the strength of the interaction should be described in the method section.

RESPONSE: Considering that the AutoDock Vina score in molecular docking simulation is an empirical binding free energy adjusted to reproduce enzyme inhibitory activity (Trott O et al.), the strength of the enzyme inhibitory activity can be predicted from the Vina scores. For example, a score of −4 kcal/mol corresponds to 100 μM and −5 kcal/mol corresponds to 10 μM, which suggests weak binding, especially for TMPRSS2. We have added the following explanation on P13 of the revised manuscript.

P13, lines 257–259: Considering that the AutoDock Vina score is an empirical binding free energy, we expected that a score of −6 kcal/mol would theoretically present the single-digit μM binding affinity with both target proteins.

However, owing to the overemphasis on “weakly,” we have removed unnecessary portions to improve the readability of our manuscript. We thank you for your comments, which have helped improve the manuscript.

Reference: Trott O, Olson AJ. AutoDock Vina: Improving the speed and accuracy of docking with a new scoring function, efficient optimization, and multithreading. J Comput Chem. 2010;31(2):455-61. Epub 2009/06/06. doi: 10.1002/jcc.21334. https://www.ncbi.nlm.nih.gov/pmc/articles/PMC3041641/

#3 In Figure 2 and Figure 4

The names of the candidate ingredients should be listed in the figure, as shown in Figure 6.

RESPONSE: Following your comment, the names of the ingredients have been added in Figures 2 and 4.

#4　　In table 3

The title of the table is "Vina score of test ingredients for human ACE2", but it should be "Vina score of test ingredients for human ACE2 model" as in Table 4.

RESPONSE: Thank you for your comments regarding the title of the table. For ACE2, as we used the X-ray crystal structure (not a model), we did not add “model” to the title of Table 4 and retained “Vina score of test ingredients for human ACE2.” However, for TMPRSS2 in Table 4, we used the homology model obtained from SWISS-MODEL (PDB ID: 5CE1), so the title was “Vina score of test ingredients for human TMPRSS2 model.”

Response to Reviewer #2:

Reviewer #2: Tateyama-Makino et al. reported the actions of ingredients in toothpaste and mouthwash expected to prevent the SARS-CoV2 infection in the oral cavity. They analyzed the inhibitory effects on the interaction between the receptor-binding domain of spike protein of SARS-CoV2 and the host receptor human ACE2, as well as human TMPRSS2, which is necessary for the viral entry to the host cells. They confirmed the effect by a docking study of the ingredients showing the high inhibitory effects to the models of human ACE2 and TMPRSS2.

Since the salivary glands and mucosae in the oral cavity are considered to be significant points for SARS-CoV-2 infection and transmission, the study could have potential importance for COVID-19 prevention. However, there are several serious concerns found in the study and the conclusion is not sufficiently supported by the data, as follows.

RESPONSE: We wish to express our appreciation to the reviewer for their insightful comments and suggestions on our manuscript which have helped us significantly improve it. We have addressed the reviewer’s comments as detailed below.

1. Surfactants in toothpaste and mouthwash are expected to affect the envelope of the virus, and thus the action might serve as the primary effect for prevention of virus infection. However, this basic effect is not discussed in the manuscript. Therefore, the significance of the actions to ACE2 and TMPRSS2, if any, among the total effects is unclear.

RESPONSE: Per your comment, surfactants are known to inactivate enveloped viruses, including SARS-CoV-2 (Simon M et al.). This description has been added to the Introduction in the manuscript. However, to further reduce the risk of virus infection, we thought that it would be effective to increase the inactivation ability of the ingredients against the virus and block the infection mechanism of the virus to inhibit its entry into the host cell. However, the effects of toothpaste and mouthwash ingredients on the infection mechanism of SARS-CoV-2 remain unrevealed. Therefore, the purpose of this study was to investigate the ingredients that affect ACE2 and TMPRSS2, which are key factors on the surface of the host cell membrane that are involved in SARS-CoV-2 infection. Considering this information, we have added the following to the Introduction in the revised manuscript:

P3, lines 56–59: Povidone-iodine and surfactants, including those found in toothpastes and mouthwashes, are known to inactivate enveloped viruses, including SARS-CoV-2 [9, 10]. To further enhance the inhibitory effect of these ingredients on the entry of SARS-CoV-2 into host cells, it is also important to consider its effect on the virus’ infection mechanism on the host cell side.

Reference: Simon M, Veit M, Osterrieder K, Gradzielski M. Surfactants - Compounds for inactivation of SARS-CoV-2 and other enveloped viruses. Curr Opin Colloid Interface Sci. 2021;55:101479-. Epub 2021/06/12. doi: 10.1016/j.cocis.2021.101479. PubMed PMID: 34149296. 

https://www.sciencedirect.com/science/article/pii/S1359029421000637

2. As the authors wrote in the Discussion, the inhibitory effects of the ingredients tested in this study are most probably derived from the denaturation of the target proteins, which are induced by non-specific binding of the ingredients to the targets. Nevertheless, the authors performed the docking study, a methodology assuming specific binding, and discussed the action based on the models. However, a docking study itself does not serve as evidence for specific binding to the targets, and should be performed based on other experimental results indicating specific binding. Nevertheless, no such evidence was provided in the manuscript. In addition, the apparent reduction of the activities might also be caused by the denaturation of the probe proteins for detection in the assay kit, such as HRP.

RESPONSE: We performed the molecular docking simulations to confirm our speculation for the inhibitory mechanisms of these ingredients. The results of molecular docking simulations in this study showed a common docking mode for all ingredients, including known inhibitors. We consider that this result supports specific binding. Based on the above, the following text was added to the manuscript:

P14, lines 265–267: The docking modes of these test ingredients were similar for the respective target proteins (Fig 6A and 6B, and S2 Fig). These results suggest that these ingredients may function through the same underlying mechanism.

Next, we would like to respond to your comment that the ingredients may have denatured the probe protein of this assay. In the in vitro assay of spike protein–ACE2 interaction in this study, the ingredients, including surfactants, were removed and washed before adding the probe protein HRP. Therefore, it is highly unlikely that these ingredients denatured the probe protein HRP or inhibited the color development. 

3. Rationales of model selection for docking studies are unclear. Especially, although the authors described that a crystal structure of human TMPRSS2 has never been resolved, the structure with an inhibitor was released on Apr. 21, 2021 (PDB ID: 7MEQ).

RESPONSE: Thank you for your comment and suggesting the reference detailing the latest information on the crystal structure of TMPRSS2 with an inhibitor. The X-ray crystal structure of human TMPRSS2 with an inhibitor had not been resolved at the time of submission of our manuscript to bioRixv (Mar 19, 2021, doi: https://doi.org/10.1101/2021.03.19.435740). Nevertheless, we have now conducted an additional analysis using the crystal structure of human TMPRSS2 (PDB ID: 7MEQ). As a result, we confirmed that our TMPRSS2 prediction model (PDB ID: 5CE1) reproduced the actual crystal structure of human TMPRSS2 (PDB ID: 7MEQ), as shown in S1 Fig of the Supporting Information, and that the docking results obtained using AutoDock Vina are reliable. We also performed molecular docking simulations using the TMPRSS2 crystal structure (PDB ID: 7MEQ). As a result, we confirmed that the results obtained using the TMPRESS2 prediction model (PDB ID: 5CE1) and the crystal structure (PDB ID: 7MEQ) are consistent, as shown in S2 Fig and S2 Table. However, unlike our TMPRSS2 prediction model (PDB ID: 5CE1), the crystal structure (PDB ID: 7MEQ) did not have the Gln438 side-chain atoms, which are present in inhibitor-binding sites. Considering that our TMPRSS2 prediction model (PDB ID: 5CE1) well reproduced the crystal structure (PDB ID: 7MEQ) (S1 Fig), we have consistently used the data of the TMPRSS2 prediction model (PDB ID: 5CE1) in the manuscript (Fig 6). Nevertheless, the data of the crystal structure (PDB ID: 7MEQ) was added as Supporting Information (S2 Fig and S2 Table). We have accordingly revised the following sections of the manuscript:

P7, lines 119–121: Conversely, the X-ray crystal structure of human TMPRSS2 had not been resolved at the time of submission of our manuscript to bioRixv.

P7, lines 124–129: As the X-ray crystal structure of human TMPRSS2 (PDB ID: 7MEQ; resolution 1.95 Å; Beldar et. al., 2021) became available when our manuscript was under review, we also prepared another human TMPRSS2 structure based on this crystal structure (PDB ID: 7MEQ) using SWISS-MODEL to refine the missing residues and atoms (side-chain rotamer of the missing Gln438 in the inhibitor-binding site was determined using the energy calculations of HMHC).

P12, lines 221–222: The X-ray crystal structure of human TMPRSS2 had not been resolved at the time of submission of our manuscript to bioRixv.

P12–13, lines 229–238: While our manuscript was under review, the X-ray crystal structure of human TMPRSS2 with nafamostat became available. Upon assessment of our TMPRSS2 model against the crystal structure, we found that our TMPRSS2 model was very similar to the crystal structure (root-mean-square deviation was 0.665Å for Ca atoms in the whole structure) and that the nafamostat binding site (fragment structure of nafamostat was covalently bound to Ser441 of the inhibitor-binding site) was identical to the inhibitor-binding site of the model. All side-chain and main-chain conformations in the inhibitor-binding site of our model showed excellent agreement with those of the human TMPRSS2 crystal structure, as shown in S1 Fig. This strongly supported the results of the docking simulations using the predicted TMPRSS2 model. We also performed the docking simulations using the refined human TMPRSS2 crystal structure (PDB ID: 7MEQ).

P14, lines 263–265: The docking simulations using the refined human TMPRSS2 crystal structure (PDB ID: 7MEQ) gave a similar AutoDock Vina score and docking mode for these test ingredients (S2 Fig and S2 Table).

Response to Reviewer #3:

Reviewer #3: In this manuscript, the authors found that general ingredients of toothpastes and mouthwashes possess inhibitory effects on the SARS-Cov2 spike protein-ACE2 interaction and the TMPRSS2 protease activity. Molecular docking study also revealed that the promising ingredients could bind to host factors at the inhibitor-binding site. The authors present some interesting possibilities that oral care products containing effective ingredients are able to reduce the viral load in the oral cavity of SARS-CoV-2 infected individuals. However, it is difficult to conceive that the experimental systems presented here is a physiological model. Additional experiments would be required to draw convincing conclusions in this study. The following comments are provided for the authors consideration.

RESPONSE: We wish to express our appreciation to the reviewer for their insightful comments and suggestions on our manuscript which have helped us significantly improve it. We have addressed the reviewer’s comments as detailed below.

Specific comments:

1. The evidence for the inhibitory effects of toothpaste and mouthwash ingredients against SARS-CoV-2 infection seems largely circumstantial, and conclusions are poorly supported, as concerned by the authors. At least, the authors should confirm that viral infectivity on the epithelial cells, such as HEK-293T overexpressing the human ACE2, Vero E6, Calu-3, is reduced in the presence of the dental care ingredients, using SARS-CoV-2 protein pseudovirus system. Addressing this comment would strengthen the conclusion.

RESPONSE: Thank you for your helpful remarks. Although this study was only validated in vitro, without the use of viruses and cells, we hope to publish these results in an open-access, peer-reviewed journal such as PLOS ONE as soon as possible. Further, we would like to leave the evaluation of viral infectivity using the SARS-CoV-2 protein pseudovirus system for future work. The reason for this is that this paper is the first paper to clarify the effectiveness of toothpaste and mouthwash ingredients on host factors involved in SARS-CoV-2 infection, and we believe that the early publication of this research in an open-access, peer-reviewed journal will further accelerate research on the prevention of SARS-CoV-2 infection in the field of oral science. We agree that the evaluation of viral infectivity using the SARS-CoV-2 protein pseudovirus system is indeed important and should be conducted in future studies. Accordingly, we have added the following to the Discussion in the manuscript. 

P20, lines 383–389: Therefore, we propose an anti-SARS-CoV-2 experiment using oral mucosa and upper respiratory tract cells and a virus infectivity evaluation test using epithelial culture cells overexpressing human ACE2 and TMPRRSS2 and pseudovirus expressed SARS-CoV-2 spike protein. Furthermore, clinical trials using these ingredients or toothpastes and mouthwashes containing these ingredients would help in further elucidating the antiviral activities of these ingredients and oral care products containing these ingredients against SARS-CoV-2 infection.

2. Figs. 1-4: The inhibitory effects of toothpaste and mouthwash ingredients diluted in PBS were evaluated in this study, which may significantly differ from a complex environment of the human oral cavity. It would be informative to see if the saliva components, such as enzymes and serum proteins, effects on the interaction between these ingredients and ACE2 or TMPRSS2.

RESPONSE: Thank you for your insightful comments on improving the quality of our manuscript. We additionally evaluated the inhibitory effects of toothpaste and mouthwash ingredients on spike protein–ACE2 interaction and TMPRSS2 protease activity in the presence of human saliva and added the obtained results to the Supporting Information (S3 Fig and S4 Fig). We evaluated the inhibitory effect of these ingredients in the presence of 10%(v/v) saliva because the maximum concentration of saliva that could be included in these assays was 10%(v/v) and because previous in vitro experiments evaluating the effects of saliva (evaluation in cell culture systems and evaluation of enzymes present in saliva) were conducted with 10%(v/v) saliva (Srinivasulu N et al., Müller HD et al., Rao BSB et al.). The results showed that the toothpaste and mouthwash ingredients (SDS, TDS, LMT, LSS, GCU, TXA, and AHA) had inhibitory effects on the binding of spike protein to ACE2 and TMPRSS2 activity in the presence of human saliva, and these inhibitory effects tended to be comparable to those observed in the in vitro assay without saliva. Please refer to Supporting Information (S3 Fig and S4 Fig) for further details. The following sentence has been added to the end of the paragraph discussing the oral effects in the manuscript:

P17, lines 312–317: In addition, the inhibitory effects of these ingredients (SDS, TDS, LMT, LSS, GCU, TXA, and AHA) on spike protein–ACE2 interaction and TMPRSS2 serine protease activity in the presence of saliva tended to be comparable to those in the absence of saliva (S3 Fig. and S4 Fig.). These results suggest that these ingredients may have an inhibitory effect on the spike protein–ACE2 interaction and TMPRSS2 serine protease activity even in the oral cavity. 

References:

Srinivasulu N, Mallaiah P, Sudhakar G, Rao BSB and Kumari DS. Alpha-amylase inhibitory activity and in vitro glucose uptake in psoas muscle and adipose tissue of male Wistar rats of leaf methanolic extract of Achyranthes aspera. Journal of Pharmacognosy and Phytochemistry 2016;5(3):176-80. https://www.phytojournal.com/archives?year=2016&vol=5&issue=3&ArticleId=871

Müller HD, Caballé-Serrano J, Lussi A, Gruber R. Inhibitory effect of saliva on osteoclastogenesis in vitro requires toll-like receptor 4 signaling. Clinical Oral Investigations 2017;21(8):2445-52. doi: 10.1007/s00784-016-2041-7.

https://link.springer.com/article/10.1007/s00784-016-2041-7

Rao BSB, Srinivasulu N, Sudhakara G, Mallaiah P, RameshB, Kumari DS. Effect of Sesbania grandiflora methanolic leaf extract on in vitro studies of α-amylase, glucose uptake in muscle and adipose tissue of male Sprague Dawley rat model. International Journal of Research and Analytical Reviews 2018;5(3):276-80. http://ijrar.com/upload_issue/ijrar_issue_1859.pdf

3. Are the ingredients effective in preventing infections with the SARS-CoV-2 variants? The authors should address this point in their discussion.

RESPONSE: Thank you for your insightful question. We consider that the ingredients found in this study are also effective in preventing prevent infections with the SARS-CoV-2 variants. We have added the following to the Discussion in the manuscript on this point:

P19–20, lines 363–378: Currently, various SARS-CoV-2 variants have emerged and are attracting attention as a cause of the spread of the infection. The major mutants [Alpha (B.1.1.7), Gamma (P.1), and Delta (B.1.617.2)], as of August 2021, have characteristic substitutions such as E484K, N501Y, D614G, or L452R in the spike protein. These substitutions do not change the binding site of ACE2 to the spike protein but rather increase the affinity between the spike protein and ACE2 [43-47]. Unlike neutralizing antibodies against SARS-CoV-2 spike protein, ingredients found in this study (SDS, TDS, LMT, LSS, and GCU) have inhibitory effects on ACE2, so it is considered that they have inhibitory effects on these SARS-CoV-2 variants as well. In addition, the ingredients found in this study (SDS, TDS, LMT, LSS, GCU, and TXA) have been shown to inhibit the serine protease activity of TMPRSS2, which is involved in the viral infection of cells after ACE2 and spike protein binding (Fig 3,4,6 and Table 4). Therefore, even if the spike protein in these SARS-CoV-2 variants binds to ACE2, these ingredients (SDS, TDS, LMT, LSS, GCU, and TXA) will inhibit TMPRSS2-dependent cell membrane fusion and thereby prevent infection of these SARS-CoV-2 variants to host cells. Considering this information, the selected toothpaste and mouthwash ingredients in this study (SDS, TDS, LMT, LSS, GCU, and TXA) may have an inhibitory effect on the infection with SARS-CoV-2 variants in the oral cavity.

References：

43. Tchesnokova V, Kulakesara H, Larson L, Bowers V, Rechkina E, Kisiela D, et al. Acquisition of the L452R mutation in the ACE2-binding interface of spike protein triggers recent massive expansion of SARS-Cov-2 variants. bioRxiv. 2021. Epub 2021/03/25. doi: 10.1101/2021.02.22.432189. https://www.biorxiv.org/content/10.1101/2021.02.22.432189v2

44. Ali F, Kasry A, Amin M. The new SARS-CoV-2 strain shows a stronger binding affinity to ACE2 due to N501Y mutant. Medicine in Drug Discovery. 2021;10:100086. doi: https://doi.org/10.1016/j.medidd.2021.100086.

45. Tian F, Tong B, Sun L, Shi S, Zheng B, Wang Z, et al. Mutation N501Y in RBD of spike protein strengthens the interaction between COVID-19 and its receptor ACE2. bioRxiv. 2021:2021.02.14.431117. doi: 10.1101/2021.02.14.431117. https://www.biorxiv.org/content/10.1101/2021.02.14.431117v2

46. Pavlova A, Zhang Z, Acharya A, Lynch DL, Pang YT, Mou Z, et al. Critical interactions for SARS-CoV-2 spike protein binding to ACE2 identified by machine learning. bioRxiv. 2021:2021.03.19.436231. doi: 10.1101/2021.03.19.436231. https://www.biorxiv.org/content/10.1101/2021.03.19.436231v1

47. Ozono S, Zhang Y, Ode H, Sano K, Tan TS, Imai K, et al. SARS-CoV-2 D614G spike mutation increases entry efficiency with enhanced ACE2-binding affinity. Nature Communications. 2021;12(1):848. doi: 10.1038/s41467-021-21118-2. https://www.nature.com/articles/s41467-021-21118-2

4. It is not clear from the Methods how the quantification of IC50 was carried out.

RESPONSE: Thank you for your comment. We have added the following information to the Materials and Methods in the manuscript to describe the method in detail:

P6, lines 92–98: The dose–response relationship between inhibition (%) and test ingredient concentration was plotted and used to determine the half maximal inhibitory concentration (IC50) using the DRC package in R software program (v3.6.1), as described in [11]. In brief, the dose–response data were fitted using a four-parametric log-logistic model or a four-parameter Brain-Cousens model when the dose–response data represented a sigmoid curve or hormesis, respectively. In addition, the obtained data was used for the estimation of the IC50.

P7, lines 110–114: The dose–response relationship between inhibition (%) and test ingredient concentration was plotted and used to determine the half maximal inhibitory concentration (IC50) using the DRC package in R software program (v3.6.1), as described in [11]. In brief, the dose–response data were fitted using a four-parametric log-logistic model and used estimate the IC50.

References:

11. Ritz C, Baty F, Streibig JC, Gerhard D. Dose-Response Analysis Using R. PLOS ONE. 2016;10(12):e0146021. doi: 10.1371/journal.pone.0146021.

Minor points

1. Fig. 1: IBE and CPB are not listed in Table 1.

RESPONSE: Thank you for pointing out this error. The abbreviations in Table 1 are correct, but those in Fig. 1 are not. IBE is incorrect, and it should instead be SCP. CPB is incorrect, and it should instead be SCP. We have revised Fig. 1 accordingly.

2. Fig. 6: The quality of model drawings is too low. It is difficult to evaluate.

RESPONSE: Following your comment, the resolution of Figure 6 has been changed from 300 dpi to 600 dpi. 

Response to Editor’s Email

RESPONSE: Thank you for recommending the deposition of our laboratory protocols in protocols.io. As this research is related to SARS-CoV-2, we hope to publish the results of this paper in a peer-reviewed, open-access journal as soon as possible. We believe that the preparation of the document for the deposition of our laboratory protocols in protocols.io would require substantial time. Hence, we would like to consider this after this manuscript is accepted.

RESPONSE: Following the instructions, we have confirmed using PACE that the figures meet the requirements of PLOS ONE.

---

## [Editor Report · Decision Letter 1]

8 Sep 2021

The inhibitory effects of toothpaste and mouthwash ingredients on the interaction between the SARS-CoV-2 spike protein and ACE2, and the protease activity of TMPRSS2 in vitro

PONE-D-21-20049R1

Dear Dr. Tateyama-Makino,

We’re pleased to inform you that your manuscript has been judged scientifically suitable for publication and will be formally accepted for publication once it meets all outstanding technical requirements.

Kind regards,

Etsuro Ito

Academic Editor

PLOS ONE